# Blue Shark (*Prionace glauca*) Distribution in the Pacific Ocean: A Look at Continuity and Size Differences

Weiwen Li [1,2], Xiaojie Dai [2,3], Kevin W. Staples [4], Bin Chen [1], Hao Huang [1] and Siquan Tian [2,3,*]

1    Laboratory of Marine Biology and Ecology, Third Institute of Oceanography, Ministry of Natural Resources, Xiamen 361005, China
2    College of Marine Sciences, Shanghai Ocean University, Shanghai 201306, China
3    Scientific Observing and Experimental Station of Oceanic Fisheries Resources and Environment, Ministry of Agriculture, Shanghai 201306, China
4    School of Marine Sciences, University of Maine, Orono, ME 04469, USA
*    Correspondence: sqtian@shou.edu.cn

**Abstract:** Blue shark (*Prionace glauca)* is a major bycatch species in the long-line and gill-net Pacific Ocean tuna fisheries, and the population structure is critical for fishery management. We employed generalized additive models to analyze the fork lengths of blue sharks and biological data (i.e., feeding level, sex, and genetic data), as well as environmental and spatial variables (i.e., sea surface temperature, month, longitude, and latitude) collected from 2011 to 2014 by the Chinese *Thunnus alalunga* long-line tuna fishery observer program. Fork length was significantly affected ($p < 0.05$) with location (latitude and longitude) and sex, and positively effected with sea surface temperature. No relationships were found between fork length and feeding level, month, and genetic data. We detected fork length differences among blue sharks over the range of the observed data, but the genetic data implied a panmictic population. Thus, we hypothesize that the genetic similarity was so close that it could not be well separated. Based on the precautionary principle, we recommend that the blue shark in the Pacific Ocean should be managed as two independent populations to ensure its sustainable use.

**Keywords:** blue shark *Prionace glauca*; distribution pattern; generalized additive model; Pacific Ocean; population structure



## 1. Introduction

Blue shark (*Prionace glauca*; Carcharhiniformes, Carcharhinidae) is considered the most abundant species of large shark worldwide, with a widespread distribution in tropical and temperate waters [1]. Blue shark is an important bycatch in tuna long-line fisheries in the Pacific Ocean, and a decrease in populations in the Western and Central Pacific Ocean during 2009 to 2014 [2] raised concerns about depletion and the possible loss of apex predators due to overfishing [3,4]. Previous studies suggest that there has been a 60% decline in the catch-per-unit-effort (CPUE) for blue sharks in the Northwest Atlantic over the past 15 years [5], and the blue shark standardized CPUE has decreased in the Pacific Ocean from 2005 to 2009 [6]. These data, along with moderate decreasing trends in the Northwest Atlantic Ocean [5], Pacific Ocean [6], and Indian Ocean [7], suggest that blue shark stocks are vulnerable. However, the latest stock assessment for blue sharks in the North Pacific indicated that production was near the maximum sustainable yield [8], calling for scientists to pay more attention to population structure, stock assessment and management recommendations.

Information about fish population structure is critical for fisheries stock assessments and management [9]. Traditional tagging experiments conducted in the Eastern, Central, and Western North Pacific Ocean provided evidence of widespread movement by blue shark throughout the North Pacific Ocean [10], but tagging data have not demonstrated

movement across the Equator in the Pacific Ocean [10,11], thereby leading to the assumption of two separate independent blue shark stocks in the Pacific Ocean [8]. However, Taguchi [9] proposed only a single stock in the Pacific Ocean based on mitochondrial DNA analysis, which is possible given that highly migratory and broadly distributed oceanic shark species often exhibit little population heterogeneity [12]. Previous population genetic studies of *Isurus oxyrinchus* [13], *Cetorhinus maximus* [14], *Rhincodon typus* [15], and *Prionace glauca* [9,16] found little to no genetic structuring among ocean basins.

Adult blue sharks are broadly distributed, whereas young blue sharks are found in the productive high latitudes of the Subtropical Convergence containing abundant prey [17,18] (Figure 1). The immature blue shark females in the Northeast Pacific Ocean are likely to move largely northward of 33° N, whereas the males move southward of 35° N during summer [19,20] because of their different temperature preferences [19]. Furthermore, the estimated growth rate (K) of male adults and juveniles combined is 0.117, whereas that for females is 0.146 [21]. The changing growth rates for males and females increase the difficulty of identifying populations using morphological methods.

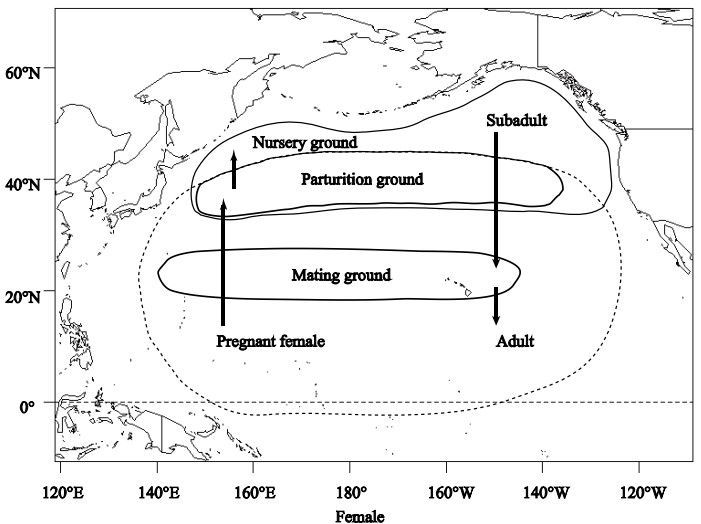

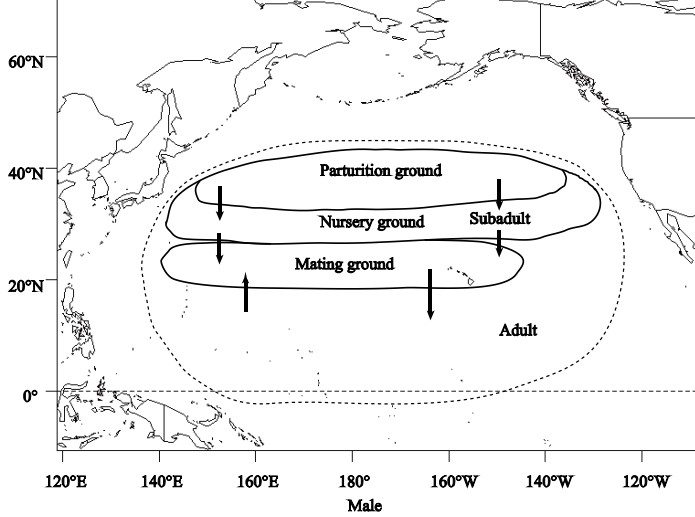

**Figure 1.** Life stages and sex segregation in North Pacific Ocean [22].

The present study uses a new dataset in order to examine population structure of blue shark population(s) in the Pacific Ocean. This is necessary because previous studies obtained conflicting results. A genetic study concluded there was a single Pacific population [9], while tagging data identified two distinct populations divided by an equatorial boundary [10,11]. In order to understand population structure in the Pacific Ocean, the generalized additive models (GAMs) were used to analyze the relationships between the biological characteristics and genotypes of blue sharks in the Pacific Ocean, and their relationships with environmental factors. The genetic differentiation of populations ($F_{ST}$) was analyzed using new genetic data to identify the blue shark population structure. Our results would like to provide scientific recommendations for sustainable fisheries and appropriate management of the species.

## 2. Materials and Methods

### 2.1. Ethics Statement

All of the blue shark samples used in this study were collected by the Chinese long-line tuna fishery observer program under the auspices of the Ministry of Agriculture for commercial fisheries (under the No.2011-08-013, No.2011-09-017, No.2012-08-011, No.2012-10-023, No.2013-08-009, No.2013-10-020, No.2014-01-002; No.2014-02-008). Permission and approval to collect and use samples were given by the Ministry of Agriculture of China. And the experiments performed in this study was under the guild of Ethics Science Committee of Shanghai Ocean University.

### 2.2. Survey Description

All data analyzed in this study were collected by the Chinese long-line fishing observer program for the fishery that targets tuna in the Pacific Ocean (including the Western and Central Pacific Ocean, and Eastern Pacific Ocean) from 2011 to 2014 (Figure 2). In total, 2340 long-line sets were observed over 4 years and biological, bycatch and fishing information data was collected onboard by trained scientific observers.

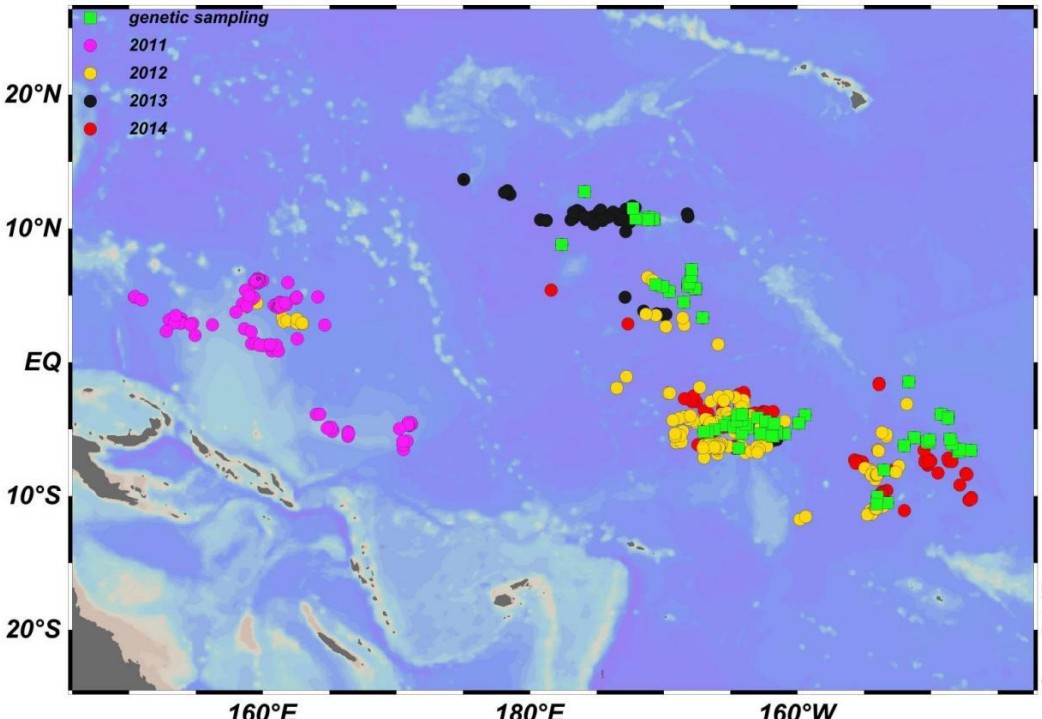

**Figure 2.** Geographic locations of the longline where onboard observers collected the data that was used in this paper.

### 2.3. Biological Data Collected

Sex, fork length (FL), and feeding level were recorded for each shark by analyzing a total of 2340 long-line sets. FL (measured from the snout to the deepest point of the tail fork) were measured to the nearest centimeter (cm). The feeding level was identified according to the methods of Fishery Resource Science [23], where the level ranged from empty stomachs (0) to full stomachs/intestines (5). And the shark were found with a pair of claspers, we determine it as male, whlie without claspers, we determine it as female.

### 2.4. Genetic Samples Collected

Muscle tissue were collected under the back fin on the board from 98 individual blue sharks in three locations (denoted as Central Pacific Ocean Part A (CPA), Central Pacific Ocean Part B (CPB), and Central Pacific Ocean Part C (CPC)) of Pacific Ocean during 2011 to 2014. Among these individuals, 31, 34, and 33 were collected from the areas comprising 0–15° N, 165–180° W (CPA), 0–10° S, 155–165° W (CPB), and 0–10° S, 145–155° W (CPC), respectively (Figure 2).

### 2.5. Genetic Marker Design and Polymerase Chain Reaction (PCR) Amplification

The primers for amplification, i.e., BSH-COI-F (5′TATAGCCTTCCCACGAATA′3) and BSH-COI-R (5′AACACCTGTAGGAATAGCG′3), were designed according to the complete mitochondrial cytochrome oxidase subunit I gene sequence (KF356249). Two primers were synthesized by Sangon Biotech (Shanghai, China). The three location samples were amplified in a 20 µL reaction mixture containing $1 \times$ PCR buffer (20 mM KCl, 4 mM, Tris-HCl [pH 8.0], 4 mM $MgCl_2$), 0.4 mM dNTPs, 1.0 mM primers, 0.5 units Taq DNA polymerase (Takara, Otsu, Japan), and 30–50 ng DNA template. The reaction was performed with an initial denaturation step at 95 °C for 40 s, followed by 30 cycles for 30 s at 95 °C, 30 s at 56 °C, and 40 s at 72 °C, and a final extension step at 72 °C for 300 s. After amplification, the products were sequenced by Sangon Biotech (Shanghai, China).

### 2.6. Genetic Data Analysis

Blue shark mitochondrial cytochrome oxidase submit I gene sequences were aligned with CLUSTALW [24]. Analysis of molecular variance (AMOVA) and $F_{ST}$ analysis were conducted by using Arlequin 3.0, and FU and its *p*-value and τ were estimated with Arlequin 3.0 [25]. Mismatch distributions from different stocks were identified with DnaSP 4.0 [26], and a neighbor-joining tree was produced using MEGA 4.0 [27]. We selected *Carcharhinus falciformis* (KF801102), *Carcharhinus obscurus* (KC470543), and *Carcharhinus galapagensis* (JQ654714) as an outgroup, and the neighbor-joining tree built under the Kimura two-parameter model. Moreover, general dates of population expansion were estimated with the formula: $T = τ/2u$ [28], where T is the time since expansion, τ is the expansion time, and $2u = µ \times$ generation time $\times$ number of basic analysis, where µ is the mutation rate. The mitochondrial DNA COI gene mutation rate $µ = 2.38 \times 10^{-9}$ substitutions per site per year [29] and an age of maturity of about 4 years [1] in the data analysis.

### 2.7. GAM Analysis

GAMs were used to explore the relationships between the variables included in our study. GAMs allow for nonlinear relationships between the response and explanatory variables [30], and thus they were useful because a priori assumptions regarding the functional forms of the responses of our independent variables were not available. FL, feeding level (FeedL), latitude (LAT), longitude (LON), sex, month (Month), sea surface temperature (SST), and genetic data (Gen) were available from the Chinese observer program, and they were included in our models. Among the variables, longitude and latitude were fitted as a two-dimensional smoothed term, SST was fitted as a single-dimensional smoothed term, and the month, feeding level, sex, and genetic data were designated as factors. Fifteen models were compared to select the optimal model. Every model included latitude, longitude, and sex, and the remaining variables were added as factors (Table 1). GAMs were

fitted using the R statistical programming environment with the '*mgcv*' package [31], where the smoothed term was automatically calculated by the program and '*gaussian*' family was selected during the analysis. The changes in log(FL) from the average were added in the plot related to SST in the final outputs. Akaike's information criterion (AIC) [32] was calculated for each model and used to select the best model(s).

**Table 1.** Variables included in the GAM formulations and their performance results.

| Model Equation | Model Number | $D_f$ | AIC |
|---|---|---|---|
| FL~s(LON,LAT) + s(SST) + FeedL + Month + Sex + Gen | Model 1 | 16.000407 | −123.6367 |
| FL~s(LON,LAT) + s(SST) + FeedL + Month + Sex | Model 2 | 32.604487 | −1230.5702 |
| FL~s(LON,LAT) + s(SST) + FeedL + Sex + Gen | Model 3 | 13.000100 | −129.2055 |
| FL~s(LON,LAT) + s(SST) + Month + Sex + Gen | Model 4 | 12.000079 | −131.2627 |
| FL~s(LON,LAT) + FeedL + Month + Sex + Gen | Model 5 | 15.000729 | −124.6096 |
| FL~s(LON,LAT) + Sex + s(SST) + FeedL | Model 6 | 12.000188 | −128.8619 |
| FL~s(LON,LAT) + Sex + s(SST) + Month | Model 7 | 19.798894 | −1237.8019 |
| FL~s(LON,LAT) + Sex + s(SST) + Gen | Model 8 | 9.000123 | −136.7627 |
| FL~s(LON,LAT) + Sex + FeedL + Month | Model 9 | 26.060133 | −1235.1461 |
| FL~s(LON,LAT) + Sex + FeedL + Gen | Model 10 | 12.000188 | −128.8619 |
| FL~s(LON,LAT) + Sex + Month + Gen | Model 11 | 11.000774 | −132.1159 |
| FL~s(LON,LAT) + Sex + s(SST) | Model 12 | 24.377982 | **−1579.5387** |
| FL~s(LON,LAT) + Sex + FeedL | Model 13 | 18.849146 | −1235.4664 |
| FL~s(LON,LAT) + Sex + Month | Model 14 | 26.031117 | **−1580.4761** |
| FL~s(LON,LAT) + Sex + Gen | Model 15 | 8.000042 | −136.4100 |

## 3. Results

### 3.1. Population Genetics Structure and Population Expansion

In total, 682 base pairs in the segment cytochrome oxidase I gene were analyzed. AMOVA (Table 2) indicated that the percentage of variation occurring within populations was 97.84% and the fixation index was 0.02164, with no significant difference. $F_{ST}$ analysis (Table 3) found no significant genetic differences among the three sampling locations. Furthermore, the neighbor-joining tree indicated that three outgroups separated from the blue shark stock clade (Figure 3), and thus there were no significant differences among the three sample stocks.

**Table 2.** Analysis of molecular variance (AMOVA) results for blue sharks.

| Source of Variation | d.f. | Sum of Squares | Variance Components | Percentage of Variation | Fixation Index |
|---|---|---|---|---|---|
| Among population | 2 | 1.455 | 0.00935 Va | 2.16 | 0.02164 |
| Within population | 95 | 40.137 | 0.42249 Vb | 97.84 | |

**Table 3.** Pairwise $F_{ST}$ analysis results for blue sharks.

| | CPA | CPB | CPC |
|---|---|---|---|
| CPA | | | |
| CPB | 0.0122 | | |
| CPC | 0.06737 | −0.0174 | |

All samples from the three locations had negative Fu's test results and the *p*-values were significant (*p* = 0; Table 4), which indicated the occurrence of a recent population expansion. Mismatch distributions from the Southern Hemisphere stock, Northern Hemisphere stock, and all samples were examined (Figure 4), and the results also suggested the occurrence of a population expansion. In addition, the mean *τ* was calculated as 0.95 (Table 4), we estimated that the population expansion might have occurred 0.15 million years ago.

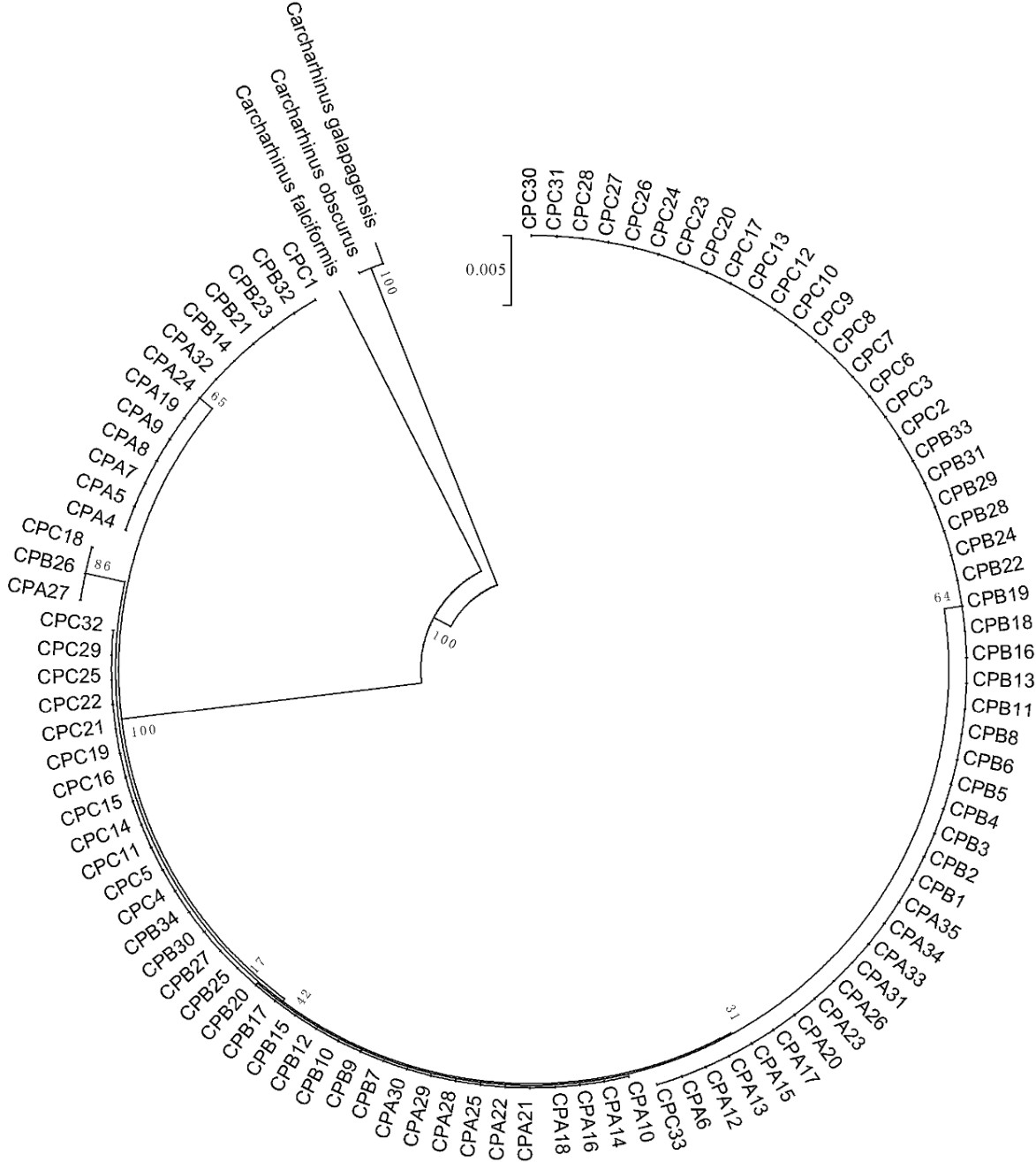

**Figure 3.** Neighbor-joining tree obtained based on 98 individual blue sharks samples from the Pacific Ocean.

**Table 4.** Parameters for mismatch distribution and Fu's test for blue sharks in the Pacific Ocean.

| Region | $\tau$ | $\theta_0$ | $\theta_1$ | FU'S | P(FU'S) |
|--------|--------|------------|------------|------|---------|
| CPA | 1.15234 | 0.00000 | 99999.00000 | <0.0000 | 0.0000 |
| CPB | 0.93359 | 0.00000 | 99999.00000 | <0.0000 | 0.0000 |
| CPC | 0.76953 | 0.00000 | 99999.00000 | <0.0000 | 0.0000 |
| Total | 0.95182 | 0.00000 | 99999.00000 | <0.0000 | 0.0000 |

<0.0000 denotes a highly negative value.

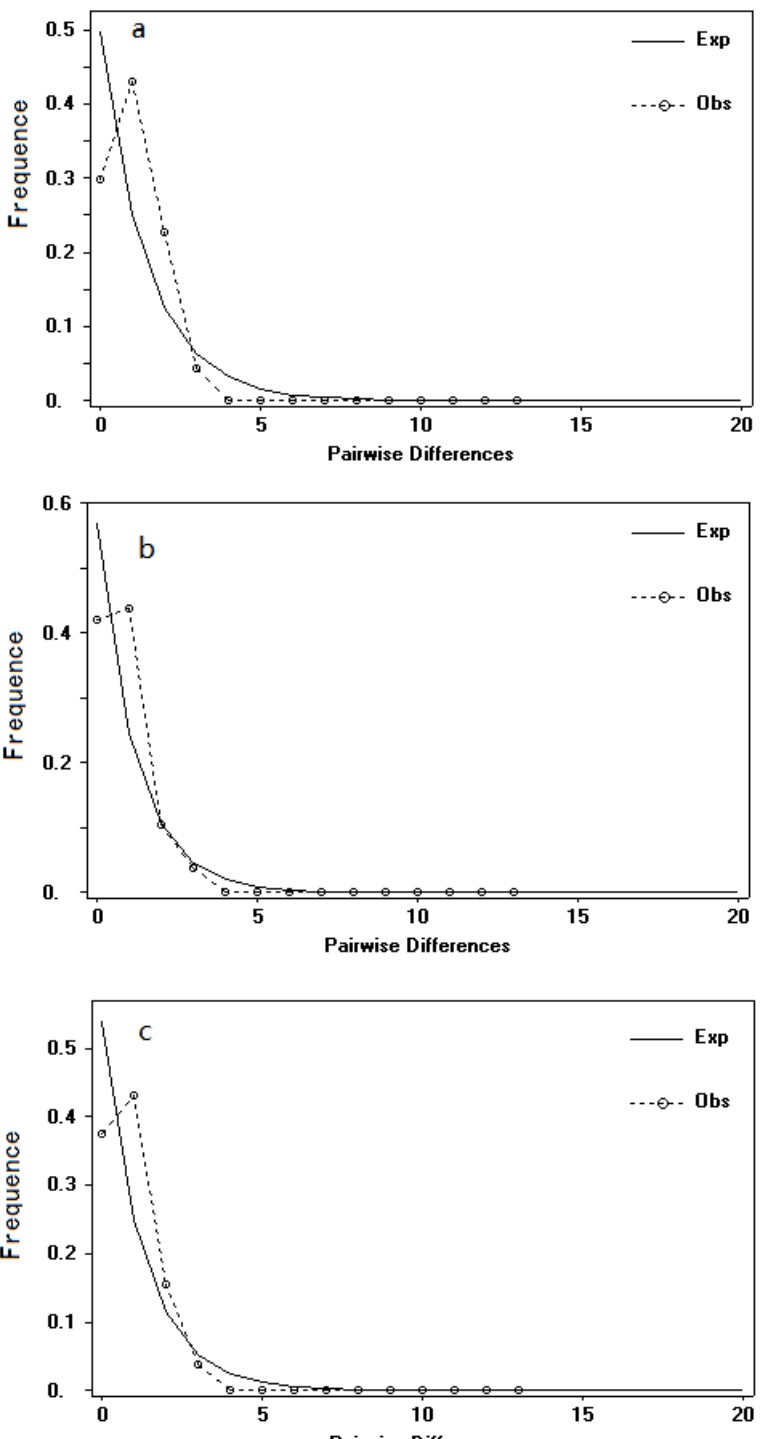

**Figure 4.** Mismatch distributions for the blue shark populations sampled in the Pacific Ocean: (**a**) mismatch distribution for the Northern Hemisphere stock; (**b**) mismatch distribution for the Southern Hemisphere stock; and (**c**) mismatch distribution for all samples. We count the number of site differences between each pair of sequences in a sample, and use the resulting counts to build the histogram.

*3.2. Modeling Results*

3.2.1. Model Selection

For the overall data set, the distribution of log(FL) was approximately normal (Figure 5). In addition, the observed cumulative probability and expected cumulative probability ap-

proximately fitted a line (Figure 6), thereby indicating that the data set was suitable for GAM analysis. Model 14 (longitude, latitude, sex, and month) obtained the lowest AIC value among the 15 models, followed by Model 12 (longitude, latitude, sex, and SST), with a slight difference in the AIC of only 1 unit. The AIC for the next closest model (Model 2) was 350 units higher (Table 1), thereby indicating that Models 12 and 14 were better at predicting the correlations between FL and the environmental variables.

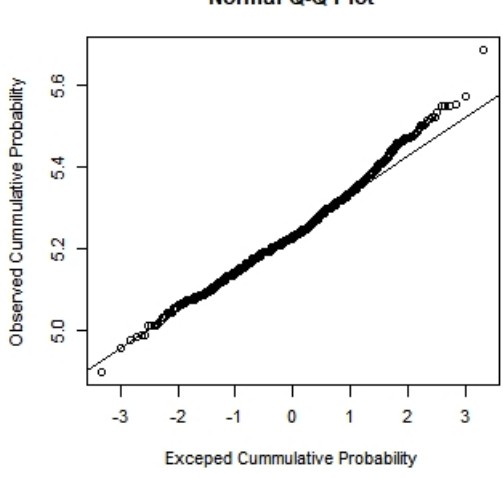

**Figure 5.** Frequency distribution for log(FL). All the FL observed were transalted into log(FL), and then the distribution were plot in the R statistical program.

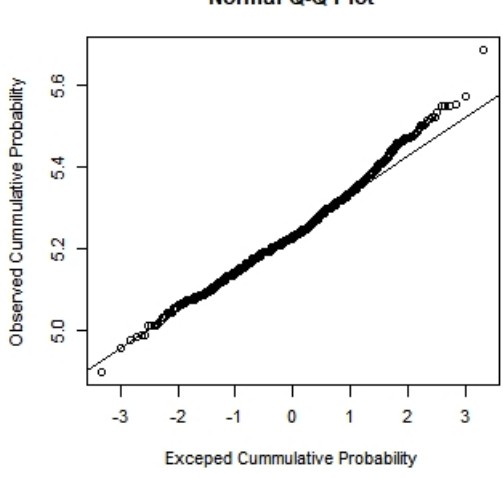

**Figure 6.** Test results for log(FL). The fit observed cummulative probability to excepted cummulative probability was used to test the distribution of log(FL) was approximately normal.

### 3.2.2. Effects of Covariates on FL in Blue Shark Individuals

Models 12 and 14 both included longitude, latitude, and sex, and each model had an additional variable (SST for Model 12 and month for Model 14). The GAMs results showed that for each model, latitude, longitude, and sex were significantly affected (i.e., $p < 0.05$) on FL (Figure 7). It indicated that FL usually increased with latitude, and it was larger in the Eastern Pacific Ocean than the Western Pacific Ocean within the study areas. Males were also larger than females. SST and month improved the AIC values for Models 12 and 14, but these variables were not significant in the GAM output (i.e., $p > 0.05$) and they were correlated. The estimated relationship of SSTand FL was roughly linear and it increased up to 29 °C, before then plateauing (Figure 7). The uncertainty around the relatively high and low SST values was very large. FL was larger in males than females (Figure 8; $t = -8.437$,

$p < 2\mathrm{e}^{-16}$). In Model 14 (April to July were not included in this study), FL was slightly larger in August and September than other months (Figure 9). Some of the highest SSTs were reported in August and September.

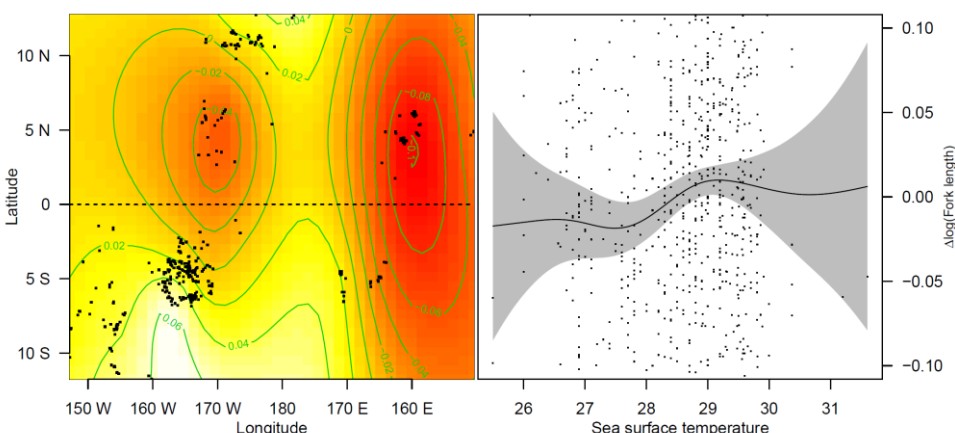

**Figure 7.** Effects of estimated environmental factors on the fork length in blue sharks in Pacific Ocean and location (**left**), and the effect of sea surface temperature on fork length (**right**, Model 12). The colors in the left-hand panel represent the changes in log (fork length) from the average, where red denotes large negative changes and white indicates large positive changes. Black dots represent the data points and contour lines indicate the change in log (fork length). In the right-hand panel, the black solid line represents the relationship estimated by the model between SST and change in log (fork length), the grey area indicates the 95% confidence interval, and the black dots are the observations.

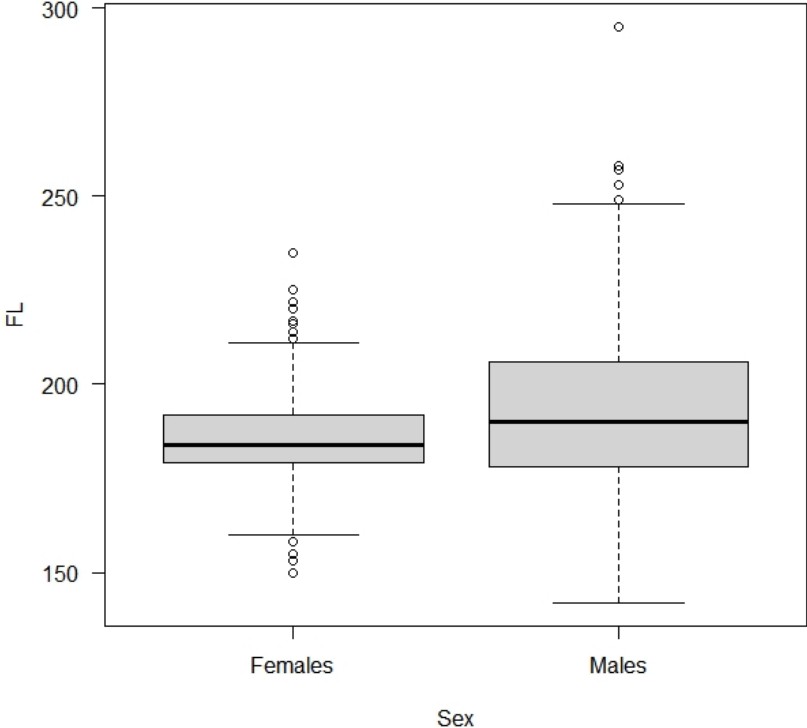

**Figure 8.** Observed fork length by sex in the Pacific Ocean.

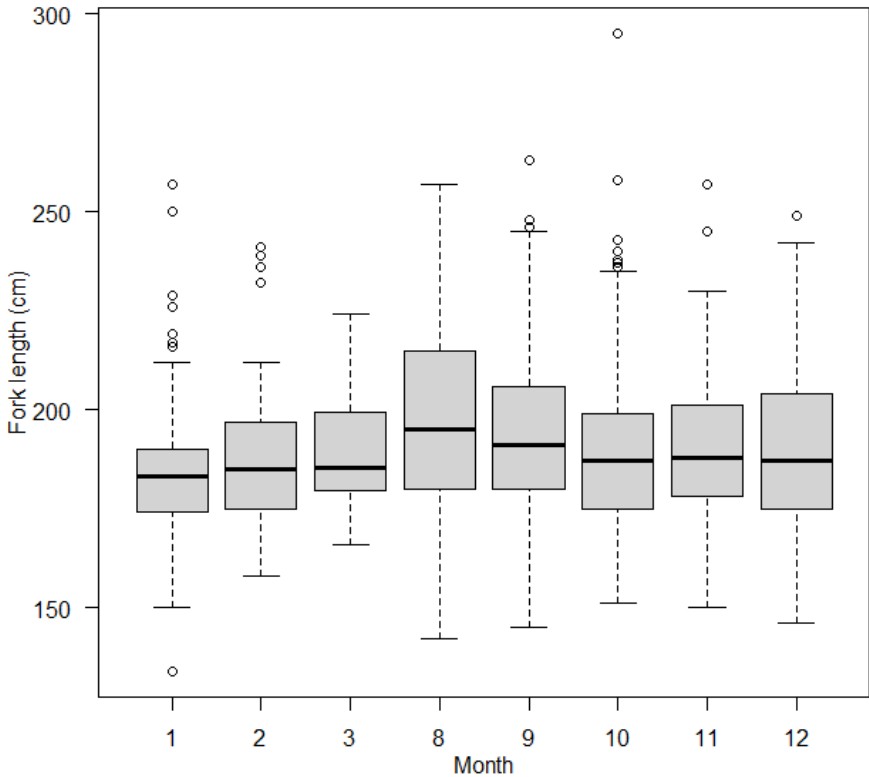

**Figure 9.** Observed fork length by month in the Pacific Ocean.

## 4. Discussion

Understanding the population structure of the blue shark in the Pacific Ocean is very important for its sustainable utilization. We found that body size of blue sharks varied according to the location and SST. No genetic differences were found in the population in the study areas in the Pacific Ocean (genetic samples were collected January, February, July, September, October, November, and December), thereby agreeing with previous studies of the blue shark's population genetic structure [9,33]. However, a "population grey zone" has been suggested that could explain a distinct single population or several independent populations [34]. Tagging studies have not demonstrated movement across the Equator in the Pacific Ocean [10,11], but these populations may be due to sex segregation and life-stage movement patterns [22,35]. Differences in morphology related to location and temperature were shown by our results, but the explanations for these differences are unknown. Ontogenetic changes in distribution and/or sex segregation by blue sharks in the Pacific Ocean are plausible explanations suggested in other studies [19,20,22,35]. High latitudes such as those north of 35° N are areas with high primary productivity and the abundant food could be beneficial for nurseries [19,20]. According to our observations, the population structure of blue sharks in the Pacific Ocean remains uncertain. However, in order to achieve the sustainable use of blue shark, we would suggest managing this species as two independent populations in the Pacific Ocean.

### 4.1. SST May Explain Body Size Differences within Sampling Areas

Our results showed that latitude, longitude, and sex were significantly affected on FL in blue sharks based on data recorded in the Pacific Ocean (Figure 7). Blue sharks in the Pacific Ocean have been shown to segregate by size throughout their development [36], which may explain the differences in body size among various sites. Thresher [37] showed that fishes inhabiting waters >250 m depth exhibited increases in body size as the temperature increased. The blue shark is a typical ectothermic species and it may exhibit changes in growth with changes in temperature. Body size and growth rate usually increase with temperature within optimal temperature ranges [38]. The optimal temperature for

blue shark growth is approximately 28.5 °C [39]. A higher temperature within the optimal temperature range may lead to increased activity and metabolism, and our findings showed that greater FL was found with higher SST values (Figure 7). Thus, a higher average SST within the optimal temperature may lead to larger body size.

Blue sharks tend to live in layers of waters with temperatures from 12–21 °C [40], where they exhibit tropical submergence to remain in the deep, cooler waters in the tropical and equatorial parts of their range [19]. However, blue sharks have been caught in oceans with SST values ranging from 8 °C to 29.5 °C [41], and the annual mean SST in the equatorial Pacific Ocean is 29 °C [42], which suggests that blue sharks should rarely be seen in the equatorial Pacific Ocean. Temperature probably affects the metabolic rate and life history processes in blue shark [43], which may explain the differences in body size found in the present study.

The Intergovernmental Panel on Climate Change assessment report [44] forecast an overall temperature increase in the surface tropical Pacific Ocean of 0.7–0.8 °C by 2035 relative to 1980–1999 and 2.5–3.0 °C by 2100 [45]. Climate change may affect the growth, body size, and distribution of marine species [46], and shifts in the ranges of many marine species have already occurred [47,48]. It is possible that climate change will influence the population dynamics of blue sharks, and thus the effects of climate change should be considered in blue shark management strategy evaluations [49]. Considering many different possible climate change scenarios will increase the robustness of management procedures under uncertainty surrounding the true changes [50], thereby providing sustainable management suggestions for managers.

### 4.2. Age-Dependent Migration and Sex Segregation May Explain Body Size Differences within Sampling Areas

Blue shark mating grounds are located in the Subtropical Convergence of the Pacific Ocean. After mating, the females generally migrate north for parturition and the juveniles linger in the northern Pacific Ocean until they are ready to mate [22]. The small juveniles grow at the Subtropical Convergence until FL reaches approximately 115 cm. Above this body size, spatial segregation of the sexes occurs by latitude and longitude. The males (FL of 70–129 cm) move southward to lower latitudes and extend their nursery area to 30° N where they remain until FL is 130–179 cm, whereas the females move northward and eastward where they remain until FL reaches 165 cm [21,51,52]. Blue sharks engage in a seasonal latitudinal migration toward tropical latitudes until they approach maturity. This special age-dependent migration and sex segregation behavior may drive the differences in morphology, and this is supported by our results because we showed that FL was significantly affected by latitude, longitude, and sex (Figures 7 and 8).

In order to effectively test this hypothesis, more representative samples in terms of age, length, sex, and location are required in the model. However, data from scientific observers were restricted to areas where the Chinese tuna fishery operated in the present study. A potential approach for addressing the gaps in this study could involve the International Scientific Committee for Tuna and Tuna-like Species collecting data uniformly from all members, and repeating a similar analysis of the population dynamics of blue sharks in the North Pacific Ocean. The results obtained would improve our knowledge of the population dynamics of blue sharks to enhance stock assessment and the management of blue sharks in the Pacific Ocean.

### 4.3. Limited Geographic Range and Sample Size for Genetic Data May Have Biased the Results

A statistically significant difference between genetic samples from two populations depends on "true" population differentiation but also the number of individuals and markers sampled [53]. Tagging data obtained by the National Oceanic and Atmospheric Administration's Southwest Fisheries Science Center, Japanese National Research Institute of Far Seas Fisheries, and New Zealand Ministry of Fisheries detected the presence of two populations of blue shark (North Pacific Ocean population and South Pacific Ocean

population) [10]. Thus, stock assessments and the management of blue sharks in the Pacific Ocean are currently separated into populations from north to south within a given ocean basin [8]. In the present study, four haplotypes were found (GenBank: KX002278–KX002281) in 98 individuals, which is fewer than the number found in other studies with larger sample sizes [9]. The sample size affects estimates of population genetic structures [54,55]. Genetic studies of school sharks (*Galeorhinus galeus*) indicated that Australian and New Zealand populations comprised a single population, but it was noted that more samples are required to confirm this finding. Furthermore, tagging data suggested a low rate of movement between the two areas [56]. Hence, the school shark populations in these two countries are assessed and managed separately [57]. As a consequence, school sharks are considered to be overfished off Australia but sustainably harvested off New Zealand, and the failure to detect populations might have contributed to this difference due to sample size limitations.

### 4.4. Divergence of the COI Gene May Be Too Recent to Identify Differences in Genetic Data

Climatic fluctuations have caused shifts in the geographic distributions of many marine species [48,58], and the effects of these shifts on population structures as well as phenotypic consequences have received much attention [59]. We estimated that population expansion might have occurred 0.15 million years ago. The population expansion occurred between the Glacial Stage and Late Glacial Stage when the temperature was 5 °C to 6 °C cooler than that at present in tropical areas [60]. Blue sharks prefer to inhabit waters with temperatures 12–21 °C [40], and thus blue sharks may have inhabited tropical waters as a single population before their recent population expansion. As the SST increased, the blue shark population may have shifted to habitats closer to the poles, which indicated a thermally adaptive response to the temperature variation [61]. Blue sharks in the Pacific Ocean might have experienced habitat shifts due to rapid temperature increases. Multiple traits dictate the potential for selective responses to these changes, which can markedly slow genetic rates of evolutionary adaptation [62]. Thus, many potential reasons may explain the absence of observed genetic differences, even when a genetically-based size difference between parts of the population(s) is apparent.

In fisheries management, the "precautionary approach" is often recommended to ensure the sustainable utilization of marine fisheries [63]. The precautionary approach stresses the importance of considering the needs of future generations and avoiding changes that are potentially irreversible. In addition, the prior identification of undesirable outcomes and measures to ensure their prompt avoidance or correction can enhance the scientific management of a fishery [64]. Stock identification is a critical component of fishery stock assessment and it is necessary for effective fishery management [65]. Misleading population identification information may lead to failures in stock assessment and management [57,62]. However, it is difficult to design scientific sampling protocols for biological and genetic studies conducted from commercially operated vessels, thereby making it hard to determine whether genetic data or biological data provide more accurate stock structures without additional information. Changes to the population structures within a stock assessment can have profound implications on the stock status and management actions. The impacts of uncertain population structures on the ability of management strategies to reach their stated goals can be examined by conducting management strategy evaluations [53]. In order to achieve sustainable fisheries usage, we recommend managing two independent blue shark populations in the Pacific Ocean.

### 5. Conclusions

In this study, we used GAM to analyze the fork lengths of blue sharks and feeding level, sex, and genetic data, sea surface temperature, month, longitude, and latitude. The results indicated that fork length was significantly correlated ($p < 0.05$) with location (latitude and longitude) and sex, and the effected with sea surface temperature were analyzed. But the relationship between fork length and feeding level, month, and genetic data are poor. And 98 individuals from 3 sampling sites were used to detected the genetic population

in the studying areas, the results implied a panmictic population. Based on the analysis by synthesis, we hypothesize that the genetic similarity was so close that it could not be well separated. Based on the precautionary principle, we recommend that the blue shark in the Pacific Ocean should be managed as two independent populations to ensure its sustainable use.

**Author Contributions:** Conceptualization, W.L. and S.T.; methodology, W.L. and X.D.; software, H.H.; validation, W.L., K.W.S., B.C. and S.T.; formal analysis, W.L. and B.C.; investigation, W.L.; data curation, B.C.; writing—original draft preparation, W.L. and X.D.; writing—review and editing, K.W.S. and S.T.; visualization, W.L.; supervision, S.T. All authors have read and agreed to the published version of the manuscript.

**Funding:** This study was supported by the Deep Sea Habitats Discovery Project of China Deep Ocean Affairs Administration (No. DY-XZ-04), Monitoring and Protection of Ecology and Environment in the East Pacific Ocean (No. DY135-E2-5-05), Monitoring and Protection of Ecology and Environment of Seamount in the Western Pacific Ocean (No. DY135-E2-2-03), China National Fishery Observer Program from Ministry of Agriculture (No. 08-54), and Shanghai Leading Academic Discipline Project (Fisheries Discipline).

**Institutional Review Board Statement:** The experiments performed in this study was under the guild of Ethics Science Committee of Shanghai Ocean University under No. SHOU-OW-2014-001 (18 May 2014).

**Data Availability Statement:** Requests to access these datasets should be directed to Dr Li. liweiwen@tio.org.cn.

**Acknowledgments:** We would like to thank the national observers who worked in the Pacific Ocean and provided data from 2009 to 2014. We thank International Science Editing (http://www.internationalscienceediting.com, accessed on 6 September 2021) for editing this manuscript.

**Conflicts of Interest:** The authors declare no conflict of interest.

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
