# Peer review of "Blue Shark (Prionace glauca) Distribution in the Pacific Ocean: A Look at Continuity and Size Differences"

_water, doi:10.3390/w15071324_

Round 1

Reviewer 1 Report

I have attached an annotated file with comments on the manuscript. While most suggestions deal with sentence rewrites and a recurring problem with affect/effect, there are other problems which need to be addressed. The use of English is actually very good with the exception of one paragraph where it all went downhill quickly, around lines 262-268.

Table 1 is mentioned in a couple of different places in the text, but it seems to refer to two different tables. The first reference seems to refer to a listing of the 98 samples used in the genetic study, but this table is not missing. I have a more comprehensive note about this in the annotated document i have attached.

The rationale for this study as explained in the introduction is to explain the population structure of blue shark using GAMs to analyze the relationship between environmental variables and genotypic data. I am not an expert in this field, i will freely admit, but i don't see that the data presented in the paper supports the confusing conclusions the authors come to that, despite morphological differences between areas, the genetic data indicated a single stock, but the authors still recommend managing as two different populations. I think the paper needs to be reworked to better explain these inconsistencies.

I am not sure that some of the figures add much to the arguments, notably figure 7. And Figure 8, according the the model statement for model 14, which the text mentions refers to Figure 8, has a sex term in the model statement, but the figure seems to be sexes combined? this confuses me.

I am afraid my judgment is that this paper needs a major rewrite, and reworking before it is ready to publish.

Author Response

Review of Blue Shark Manuscript

Annotated List of Comments

Line 20 – ‘affected’ not effected

Response: Thanks for your suggestion. We have revised as your suggestion in the text.

Line 34 – suggested rewrite ‘…in the Pacific Ocean, and a decrease in populations in the Western and Central Pacific Ocean during 2009-2014 raised concerns….’

Response: Thanks for your suggestion. We have revised as your suggestion in the text.

Line 39 – suggested rewrite – ‘These data, along with moderate decreasing trends….and Mediterranean Sea, suggest that blue shark stocks are vulnerable…”

Response: Thanks for your suggestion. We have revised as your suggestion in the text.

Line 44-delete s at end of structures

Response: Thanks for your suggestion. We have revised as your suggestion in the text.

Line 55-replace found with concentrated

Response: Thanks for your suggestion. We have revised as your suggestion in the text.

Line 57-59 Isnt northward of 33N essentially the same as southward of 35N? If the 2 sexes are occupying the same water, how can you say they have different temperature preferences?

Response:Thanks for you suggestion. The immature blue shark females in the Northeast Pacific Ocean are likely to move largely northward of 33°N, whereas the males move southward of 35°N during summer [19-20] because of their different temperature preferences [19]. Females are northward, while males are southward, they are not share the same space.

Line 59 – this whole sentence, about different growth rates, seems out of place in a paragraph whose topic sentence is about geographic distribution. Also, your last sentence (lines 61-62) talks about different growth rates during ‘life stages’ but your referring back to different sexes. Life stages are not sexes. This whole phrasing seems confusing and needs to be re-organized and rewritten.

Response:Thanks for you suggestion. We have re-organized as: ”Furthermore, the estimated growth rate (K) of male adults and juveniles combined is 0.117, whereas that for females is 0.146 [21]. The changing growth rates for males and females increase the difficulty of identifying populations using morphological methods.”

Line 63 -reorganize paragraph – suggested rewrite – ‘The present study uses a new dataset in order to examine population structure of blue shark population(s) in the Pacific Ocean. This is necessary because previous studies obtained conflicting results. A genetic study concluded there was a single Pacific population [x], while tagging data identified two distinct populations divided by an equatorial boundary [x, xx]. General additive…..’

Response: Thanks for your suggestion. We have revised as your suggestion in the text.

Line 98 – ‘ranged from empty stomachs (0) to full stomachs/intestines (5)’

Response: Thanks for your suggestion. We have revised as your suggestion in the text.

Line 101-102- It would be helpful if you could identify/label the three locations on your map figure, draw in the polygons and label? Would add to the figure.

Response: Thanks for your suggestion. We have replaced the new figure in the text.

Line 103 – Reference here to Table 1, which from your text I thought would be a listing of the 98 individual specimens, but the table in the manuscript is a listing of the 15 GAM models, something is really wrong here!

Response: Thanks for your suggestion. We have deleted (table ) in the text.

Line 105-106 – re-emphasizing that it would helpful to label the three areas on your figure .

Response: Thanks for your suggestion. We have replaced the new figure in the text.

Line 138 – another reference to Table 1 which appears to be something different than the Table 1 you mentioned previously

Response: Thanks for your suggestion. We have checked it.

Line 148 – you say ‘% of variation among populations was 97.84%’, but if I read Table 2 right, the 97.84% number belongs to the within populations line, not among……are things in the right place?

Response: Thanks for your suggestion. 97.84% number belongs to the within population, we have revised in the text.

Line 182-183- the phrase ‘observed cumulative probability and expected cumulative probability approximately fitted a line’ is confusing to me. Is that the only requirement for making data acceptable to use in a GAM model? To ‘fit a line’ could span a wide spectrum of actual fitting…..also, the cumulative probabilities are probabilities of what?

Response:Thanks for you suggestion. The distribution of data was approximately normal was acceptable to use in a GAM model, while the cumulative probabilities was used to test normal distribution of the data.

Line 188 – not sure you need Table 1 reference here since you just cited earlier in the same sentence

Response: Thanks for your suggestion. We have deleted (table 1) in the text.

Line 196 – affected, not effected

Response: Thanks for your suggestion. We have revised as your suggestion in the text.

Line 197 – Rewrite, not sure what ‘SST affected FL….’ Means

Response:Thanks for your suggestion.We have rewritten “SST effected on FL (edf = 3.383, P = 0.283), where the result indicated that FL usually increased with latitude, and it was larger in the Eastern Pacific Ocean than the Western Pacific Ocean within the study areas.” as “It indicated that FL usually increased with latitude, and it was larger in the Eastern Pacific Ocean than the Western Pacific Ocean within the study areas. ”

Line 201 – rewrite ‘The estimated effects of SST on FL…’

Response:Thanks for your suggestion.We have rewritten “The estimated of SST effected on FL was roughly linear and it increased up to 29°C, before then plateauing (Figure 6). “ as “The estimated relationship of SST and FL was roughly linear and it increased up to 29°C, before then plateauing (Figure 6).”

Line 206 – delete ‘Coincidentally’…..excess unnecessary verbiage.

Response: Thanks for your suggestion. We have revised as your suggestion in the text.

Line 201 – rewrite ‘The estimated effects of SST on FL…’

Response:Thanks for you suggestion. We rewrite as” The estimated relationship of SSTand FL was roughly linear and it increased up to 29°C, before then plateauing (Figure 6). The uncertainty around the relatively high and low SST values was very large.”

Line 209 – ‘affected’ or do you mean ‘Estimated environmental factors that have an effect on fork length….and the effect of SST on FL…..’?

Response:Thanks for you suggestion. Yes, ‘affected’ means ‘Estimated environmental factors that have an effect on fork length…..’.

Figure 7 – not sure this figure is relevant to overall point of paper

Response:Thanks for you suggestion. Figure 7 would showed the the difference of mean body size between males and females.

Line 220 – Is this sexes combined? If so, say so! Your model statement for model 14 which is identified as this figure, has a sex terms, so why does this figure appear to be sex-less, or combined sexes?

Response:Thanks for you suggestion. Figure 7 would showed the the difference fork length observed in different month from the data-set.

Line 225 – ‘…was collected January, February, July, September-December…’

Response: Thanks for your suggestion. We have revised as your suggestion in the text.

Line 245 - ‘affected’

Response: Thanks for your suggestion. We have revised as your suggestion in the text.

Line 245 – if by ‘lower than’ you mean ‘deeper’ then change to ‘…inhabiting waters >250 m depth…’

Response: Thanks for your suggestion. We have revised as your suggestion in the text.

Line 249 – delete ‘close to’ – replace with ‘approximately’ if you like, but not necessary

Response: Thanks for your suggestion. We have revised as your suggestion in the text.

Line 250 – replace ‘promote….metabolism’ with ‘lead to increased activity and metabolism…’

Response: Thanks for your suggestion. We have revised as your suggestion in the text.

Line 261 – change ‘boy’ to ‘body’?

Line 262-263 – this sentence needs major revision, I don’t know where to start

Line 263-265 – Terrible sentence construction, please rewrite!

Line 265 – ‘Sub-adults will travel from cooler to warmer areas…’

Line 266 – why will ‘it’ increase growth rates?? Just because it travelled to ‘hotter areas’?

Line 267 – male adults became larger, just because females preferred larger males?? Citation?? If you are going to make this statement you need to justify it, explain it better!

Line 267 – ‘This phenomenon is similar…’ what phenomenon? Clarify

Thanks for you suggestion. We have deleted “ Higher temperature and body mass increased the metabolic rate [66,67]. When the adult blue shark in the subtropical converge zone, it grow faster, and then travel from south to north to mating [34]. While the young blue shark state in the cooler areas to decrease its metabolic rate, reducing the consumption of organic from body to keeps gain weight. Sub-adults will travel from cooler to warmer areas to increase its growth rate. Females preferred to mate with large males [68], so the male adults became larger in the long term. This phenomenon is similar with the migration pattern of blue shark. “ from line 261 to 268.

Line 333- ‘thermally adaptive response may have occurred in terms of body size….’ What exactly does this mean? Does it mean the magnitude of movement in response to warming temps differed based on whether fish were large or small? Needs to be clarified and rewritten.\

Response:Thanks for you suggestion. We have rewritten as “As the SST increased, the blue shark population may have shifted to habitats closer to the poles, which indicated a thermally adaptive response to the temperature variation [61]”

Line 334 – ‘habitat shifts due to increased fishing pressure’?? So, you were talking about the glacial and late glacial periods, and the last glacial ended 11,700 years ago, so would fishing pressure have even been a thing then? I don’t think so, need to rewrite.

Response:Thanks for you suggestion. We have rewritten as “Blue sharks in the Pacific Ocean might have experienced habitat shifts due to rapid temperature increases. “ in the text.

Reviewer 2 Report

I has reviewed the ms (Water-2257647). The authors employed generalized additive models to analyze the fork lengths and biological data of blue sharks, as well as environmental and spatial variables. The results revealed the fork length of the fish was significantly effected with location and sex, and positively effected with sea surface temperature. And the genetic analysis indicated the fish is a panmictic population in the Pacific Ocean. Combining with previous data, the authors recommend that the blue shark in the Pacific Ocean should be managed as two independent populations to ensure its sustainable use. The study has huge workload and quite difficulty in field and has potentially interesting in management of fisheries and conservation of fish biodiversity. I provide in detail my concerns for all sections below, hoping authors will find them to be useful. Furthermore, I am not a native English speaker, so my comments will mainly not involve the issues of language.

Major concern:

Mostly, I am curious in biological characteristics of the fish which is critical to population structures of the fish, and then to fisheries stock assessments and management for the fish. The authors collected data in distribution, sex, fork length, and feeding level of the fish in sampling sites. However, the ms did not show these results. I suggested the authors added these results to provide basically biological data for further researches and management. Furthermore, the sampling size used in this study is not clear because biological results, i.e., results in distribution, sex, fork length, and feeding level, were absent in the ms. So I can't catch deeper understanding the results of the generalized additive models. Please state the sampling size to provide readers that data is not unbalanced by sampling years, sex ,or size groups.

Specific concerns:

Line 4: the indicia on the correspondence author is strange to me.

Line 11: delete : between “Correspondence: author”.

Line 22: the authors did not detected morphological differences except fork length.

Lines 24-26: the recommendation of the authors needed a justification, because the last sentence said that the sampling fish was a panmictic population and genetic similarity was so close that it could not be well separated.

Lines 57-59: it is better that the authors add migration routes of different sex and life stages in sampling map or another map. The migration routes are important to understand the findings of this study.

Lines 59-61: the sentence indicated the male individuals have lower growth rate. And the results revealed the male was bigger than the female. So age structures of specimen between sex used in this study should be different.  

Line 94: add information in determining the sex?

Line 96: delete “and clasper length”.

Line 97: add a reference for “Fishery Resource Science”.

Line 103: Table 1?

Line 111: what were the three mitochondrial genes?

Line 121: add information for FU and “τ ” when the first appearances of some abbreviations.

Lines 148-150: delete “, thereby suggesting that the variation came from the sub-population level”. the AMOVA result did not support this inference.

Lines 152-154: remove into Material and Method.

Lines 159-160: remove into Material and Method.

Lines 162-163: remove into Material and Method. These all were not results.

Table 3 and 4: add explanations for abbreviations. The explanations also need to add for other tables and figures if the issues included some abbreviations. By the way, some legends and captions in the ms were too simple, such as Table 1, Figure 4, Figure 5. The readers can't catch the useful information from the captions if they do it without text.

Lines 312-313: remove into Material and Method.

The Discussion part: the authors revealed the fork length of the fish was significantly effected with location and sex, and positively effected with sea surface temperature. And then the authors discussed roles of water temperature and age-dependent migration and sex segregation in shaping body size differences within sampling areas. So what can be recommended based on these results and discussions for management and conservation of the fish? 

Author Response

Comments and Suggestions for Authors

I has reviewed the ms (Water-2257647). The authors employed generalized additive models to analyze the fork lengths and biological data of blue sharks, as well as environmental and spatial variables. The results revealed the fork length of the fish was significantly effected with location and sex, and positively effected with sea surface temperature. And the genetic analysis indicated the fish is a panmictic population in the Pacific Ocean. Combining with previous data, the authors recommend that the blue shark in the Pacific Ocean should be managed as two independent populations to ensure its sustainable use. The study has huge workload and quite difficulty in field and has potentially interesting in management of fisheries and conservation of fish biodiversity. I provide in detail my concerns for all sections below, hoping authors will find them to be useful. Furthermore, I am not a native English speaker, so my comments will mainly not involve the issues of language.

Major concern:

Mostly, I am curious in biological characteristics of the fish which is critical to population structures of the fish, and then to fisheries stock assessments and management for the fish. The authors collected data in distribution, sex, fork length, and feeding level of the fish in sampling sites. However, the ms did not show these results. I suggested the authors added these results to provide basically biological data for further researches and management. Furthermore, the sampling size used in this study is not clear because biological results, i.e., results in distribution, sex, fork length, and feeding level, were absent in the ms. So I can't catch deeper understanding the results of the generalized additive models. Please state the sampling size to provide readers that data is not unbalanced by sampling years, sex ,or size groups.

Response: Thanks for your good suggestion.  We have provided biological data information included in the MS below.

Biological data information included in the MS

Biological Data

NO.samples

Mean FL ±SD

Sex

Males

595

192.9±21.3

Females

221

185.8±13.8

Feeding
 Level

0

403

139.6±19.1

1

110

191.5±17.4

2

82

193.9±20.1

3

33

196.7±27.9

4

21

194.6±24.4

5

0

0

Specific concerns:

Line 4: the indicia on the correspondence author is strange to me.

Response: Thanks for your good suggestion. We have deleted and in the text.

Line 11: delete “:” between “Correspondence: author”.

Response: Thanks for your good suggestion. We have deleted : in the text.

Line 22: the authors did not detected morphological differences except fork length.

Response: Thanks for your goog suggestion. We have placed “morphological” to “fork length” in the text.

Lines 24-26: the recommendation of the authors needed a justification, because the last sentence said that the sampling fish was a panmictic population and genetic similarity was so close that it could not be well separated.

Response: Thanks for your good suggestion. We have rewritten the line 24-26 as “ Thus, we hypothesize that the genetic similarity was so close that it could not be well separated. Based on the precautionary principle, we recommend that the blue shark in the Pacific Ocean should be managed as two independent populations to ensure its sustainable use.” in the text.

Lines 57-59: it is better that the authors add migration routes of different sex and life stages in sampling map or another map. The migration routes are important to understand the findings of this study.

Response: Thanks for your good suggestion. We have added figure 1 in the text.

Figure 1. stages and sex segregation in North Pacific Ocean (reproduced from Nakano, 1994).

Lines 59-61: the sentence indicated the male individuals have lower growth rate. And the results revealed the male was bigger than the female. So age structures of specimen between sex used in this study should be different.  

Response: Thanks for your good suggestion. We have cited the reference from Fujinami et al., 2019, and the data for analysis seems that the precaudal length of females is larger than males which similar with our results. And the different growth rate in the text also cited from Fujinami et al., 2019.

  1. Fujinami, Y., Semba, Y., & Tanaka, S. (2019). Age determination and growth of the blue shark (Prionace glauca) in the western North Pacific Ocean. Fishery Bulletin, 117:107–120.

Line 94: add information in determining the sex?

Response: Thanks for you good suggestion. We have added the information “And the shark were found with a pair of claspers, we determine it as male, whlie without claspers, we determine it as female.” in the text.

Line 96: delete “and clasper length”.

Response: Thanks for your suggestion. We have revised as your suggestion in the text.

Line 97: add a reference for “Fishery Resource Science”.

Response: Thanks for your suggestion. We have added the reference to the “Fishery Resource Science”.

Line 103: “Table 1”?

Response: Thanks for your good suggestion. We have deleted (table 1) in the text.

Line 111: what were the three mitochondrial genes?

Response: Thanks for your good suggestion. We have replaced the sentence as” The three location samples were amplified in a 20 L reaction mixture containing 1 × PCR buffer (20 mM KCl, 4 mM, Tris-HCl [pH 8.0], 4 mM MgCl2), 0.4 mM dNTPs, 1.0 mM primers, 0.5 units Taq DNA polymerase (Takara, Otsu, Japan), and 30–50 ng DNA template. “ in the text.

Line 121: add information for “FU” and “τ ” when the first appearances of some abbreviations.

Response: Thanks for your suggestion.“FU” and “τ ”  are conventional letter. FU used to indicate the frequency polymorphisms, while τ  is the expansion time.

Lines 148-150: delete “, thereby suggesting that the variation came from the sub-population level”. the AMOVA result did not support this inference.

Response: Thanks for your suggestion. We have revised as your suggestion in the text.

Lines 152-154: remove into Material and Method.

Response: Thanks for your suggestion. We have revised as your suggestion in the text.

Lines 159-160: remove into Material and Method.

Response: Thanks for your suggestion. We have revised as your suggestion in the text.

Lines 162-163: remove into Material and Method. These all were not results.

Response: Thanks for your suggestion. We have revised as your suggestion in the text.

Table 3 and 4: add explanations for abbreviations. The explanations also need to add for other tables and figures if the issues included some abbreviations. By the way, some legends and captions in the ms were too simple, such as Table 1, Figure 4, Figure 5. The readers can't catch the useful information from the captions if they do it without text.

Response: Thanks for your suggestion. We have revised as your suggestion in the text.

Lines 312-313: remove into Material and Method.

Response: Thanks for your suggestion. We have revised as your suggestion in the text.

The Discussion part: the authors revealed the fork length of the fish was significantly effected with location and sex, and positively effected with sea surface temperature. And then the authors discussed roles of water temperature and age-dependent migration and sex segregation in shaping body size differences within sampling areas. So what can be recommended based on these results and discussions for management and conservation of the fish? 

Response: Thanks for your good suggestion. We have achieve an recommendation in the text “ In order to achieve sustainable fisheries usage, we recommend managing two independent blue shark populations in the Pacific Ocean. “ in the line 359-361.

Comments and Suggestions for Authors

I has reviewed the ms (Water-2257647). The authors employed generalized additive models to analyze the fork lengths and biological data of blue sharks, as well as environmental and spatial variables. The results revealed the fork length of the fish was significantly effected with location and sex, and positively effected with sea surface temperature. And the genetic analysis indicated the fish is a panmictic population in the Pacific Ocean. Combining with previous data, the authors recommend that the blue shark in the Pacific Ocean should be managed as two independent populations to ensure its sustainable use. The study has huge workload and quite difficulty in field and has potentially interesting in management of fisheries and conservation of fish biodiversity. I provide in detail my concerns for all sections below, hoping authors will find them to be useful. Furthermore, I am not a native English speaker, so my comments will mainly not involve the issues of language.

Major concern:

Mostly, I am curious in biological characteristics of the fish which is critical to population structures of the fish, and then to fisheries stock assessments and management for the fish. The authors collected data in distribution, sex, fork length, and feeding level of the fish in sampling sites. However, the ms did not show these results. I suggested the authors added these results to provide basically biological data for further researches and management. Furthermore, the sampling size used in this study is not clear because biological results, i.e., results in distribution, sex, fork length, and feeding level, were absent in the ms. So I can't catch deeper understanding the results of the generalized additive models. Please state the sampling size to provide readers that data is not unbalanced by sampling years, sex ,or size groups.

Response: Thanks for your good suggestion.  We have provided biological data information included in the MS below.

Biological data information included in the MS

Biological Data

NO.samples

Mean FL ±SD

Sex

Males

595

192.9±21.3

Females

221

185.8±13.8

Feeding
 Level

0

403

139.6±19.1

1

110

191.5±17.4

2

82

193.9±20.1

3

33

196.7±27.9

4

21

194.6±24.4

5

0

0

Specific concerns:

Line 4: the indicia on the correspondence author is strange to me.

Response: Thanks for your good suggestion. We have deleted and in the text.

Line 11: delete “:” between “Correspondence: author”.

Response: Thanks for your good suggestion. We have deleted : in the text.

Line 22: the authors did not detected morphological differences except fork length.

Response: Thanks for your goog suggestion. We have placed “morphological” to “fork length” in the text.

Lines 24-26: the recommendation of the authors needed a justification, because the last sentence said that the sampling fish was a panmictic population and genetic similarity was so close that it could not be well separated.

Response: Thanks for your good suggestion. We have rewritten the line 24-26 as “ Thus, we hypothesize that the genetic similarity was so close that it could not be well separated. Based on the precautionary principle, we recommend that the blue shark in the Pacific Ocean should be managed as two independent populations to ensure its sustainable use.” in the text.

Lines 57-59: it is better that the authors add migration routes of different sex and life stages in sampling map or another map. The migration routes are important to understand the findings of this study.

Response: Thanks for your good suggestion. We have added figure 1 in the text.

Figure 1. stages and sex segregation in North Pacific Ocean (reproduced from Nakano, 1994).

Lines 59-61: the sentence indicated the male individuals have lower growth rate. And the results revealed the male was bigger than the female. So age structures of specimen between sex used in this study should be different.  

Response: Thanks for your good suggestion. We have cited the reference from Fujinami et al., 2019, and the data for analysis seems that the precaudal length of females is larger than males which similar with our results. And the different growth rate in the text also cited from Fujinami et al., 2019.

  1. Fujinami, Y., Semba, Y., & Tanaka, S. (2019). Age determination and growth of the blue shark (Prionace glauca) in the western North Pacific Ocean. Fishery Bulletin, 117:107–120.

Line 94: add information in determining the sex?

Response: Thanks for you good suggestion. We have added the information “And the shark were found with a pair of claspers, we determine it as male, whlie without claspers, we determine it as female.” in the text.

Line 96: delete “and clasper length”.

Response: Thanks for your suggestion. We have revised as your suggestion in the text.

Line 97: add a reference for “Fishery Resource Science”.

Response: Thanks for your suggestion. We have added the reference to the “Fishery Resource Science”.

Line 103: “Table 1”?

Response: Thanks for your good suggestion. We have deleted (table 1) in the text.

Line 111: what were the three mitochondrial genes?

Response: Thanks for your good suggestion. We have replaced the sentence as” The three location samples were amplified in a 20 L reaction mixture containing 1 × PCR buffer (20 mM KCl, 4 mM, Tris-HCl [pH 8.0], 4 mM MgCl2), 0.4 mM dNTPs, 1.0 mM primers, 0.5 units Taq DNA polymerase (Takara, Otsu, Japan), and 30–50 ng DNA template. “ in the text.

Line 121: add information for “FU” and “τ ” when the first appearances of some abbreviations.

Response: Thanks for your suggestion.“FU” and “τ ”  are conventional letter. FU used to indicate the frequency polymorphisms, while τ  is the expansion time.

Lines 148-150: delete “, thereby suggesting that the variation came from the sub-population level”. the AMOVA result did not support this inference.

Response: Thanks for your suggestion. We have revised as your suggestion in the text.

Lines 152-154: remove into Material and Method.

Response: Thanks for your suggestion. We have revised as your suggestion in the text.

Lines 159-160: remove into Material and Method.

Response: Thanks for your suggestion. We have revised as your suggestion in the text.

Lines 162-163: remove into Material and Method. These all were not results.

Response: Thanks for your suggestion. We have revised as your suggestion in the text.

Table 3 and 4: add explanations for abbreviations. The explanations also need to add for other tables and figures if the issues included some abbreviations. By the way, some legends and captions in the ms were too simple, such as Table 1, Figure 4, Figure 5. The readers can't catch the useful information from the captions if they do it without text.

Response: Thanks for your suggestion. We have revised as your suggestion in the text.

Lines 312-313: remove into Material and Method.

Response: Thanks for your suggestion. We have revised as your suggestion in the text.

The Discussion part: the authors revealed the fork length of the fish was significantly effected with location and sex, and positively effected with sea surface temperature. And then the authors discussed roles of water temperature and age-dependent migration and sex segregation in shaping body size differences within sampling areas. So what can be recommended based on these results and discussions for management and conservation of the fish? 

Response: Thanks for your good suggestion. We have achieve an recommendation in the text “ In order to achieve sustainable fisheries usage, we recommend managing two independent blue shark populations in the Pacific Ocean. “ in the line 359-361.

Reviewer 3 Report

This study analyzed the fork lengths of blue sharks and biological data as well as environmental and spatial variables collected from 2011 to 2014 by the Chinese Thunnus alalunga long-line tuna fishery observer program. They recommend that the blue shark in the Pacific Ocean should be managed as two independent populations to ensure its sustainable use. This paper deals with relevant topic and is potentially a valuable contribution to studies about marine fish conservation and management. However, there are following concerns.

1.     Please check typo, etc. There are minor mistakes in this manuscript.

2.     In the abstract, the authors recommend that the blue shark in the Pacific Ocean should be managed as two independent populations to ensure its sustainable use. What is your definition of “two independent populations?” 

3.     There are no hypotheses and predictions we can follow in the introduction. I recommend that the authors describe the hypotheses and predictions on this study in the introduction of this paper.

4.     The authors need to better describe the specific objectives of this study in the introduction.

5.     The authors need to provide more information about the sampling design (e.g., sampling season, effort, etc.).

6.     There are numerous nonlinear statistics in the world. The authors need to better describe why GAM is appropriate nonlinear analysis for this study. In addition, the authors need to better explain why the linear analysis is not appropriate for this study. 

7.     The discussion seems to be out of focus possibly because the specific objectives of this manuscript were not described well in the introduction. 

8.     The conservation/management suggestions are limited in this study. I recommend that the authors write the more specific conservation/management strategies based on your results.

Author Response

Comments and Suggestions for Authors

This study analyzed the fork lengths of blue sharks and biological data as well as environmental and spatial variables collected from 2011 to 2014 by the Chinese Thunnus alalunga long-line tuna fishery observer program. They recommend that the blue shark in the Pacific Ocean should be managed as two independent populations to ensure its sustainable use. This paper deals with relevant topic and is potentially a valuable contribution to studies about marine fish conservation and management. However, there are following concerns.

  1. Please check typo, etc. There are minor mistakes in this manuscript.

Response: Thanks for your good suggestion. We have checked this mistakes in the MS, and revised in the text.

  1. In the abstract, the authors recommend that the blue shark in the Pacific Ocean should be managed as two independent populations to ensure its sustainable use. What is your definition of “two independent populations?” 

Response: Thanks for your good suggestion. These two population in the abstract indicated a population form South Pacific Ocean, and another population form North Pacific Ocean based on the observed data.

  1. There are no hypotheses and predictions we can follow in the introduction. I recommend that the authors describe the hypotheses and predictions on this study in the introduction of this paper.

Response: Thanks for your good suggestion. Based on the references study, we could not predict the conclusion. Based on the data we observed, together with previous studies, then we recommended manage as two independent population. 

  1. The authors need to better describe the specific objectives of this study in the introduction.

Response: Thanks for your good suggestion. We have improved it as your suggestion in the text.

  1. The authors need to provide more information about the sampling design (e.g., sampling season, effort, etc.).

Response: Thanks for your good suggestion.  The sampling effort are based on the random sampling according to the commercial fisheries activities. Much detail of the effort are showed below.

Month

Jan

Feb

Mar

Agu

Ste

Oct

Nov

Dec

Observed indivual

174

85

40

65

159

214

198

162

  1. There are numerous nonlinear statistics in the world. The authors need to better describe why GAM is appropriate nonlinear analysis for this study. In addition, the authors need to better explain why the linear analysis is not appropriate for this study. 

Response: Thanks for your good suggestion.  Because GAMs allow for nonlinear relationships between the response and explanatory variables [28], and thus they were useful because a priori assumptions regarding the functional forms of the responses of our independent variables were not available. The relation between FL and other variables are not well known, so we use GAM for the analysis in this study.

  1. The discussion seems to be out of focus possibly because the specific objectives of this manuscript were not described well in the introduction. 

Response: Thanks for your good suggestion. The size different were found from North and South Pacific Ocean. And the SST, migration pattern, sex segregation are equal to the two population. Genetic divergence may be too recent to identify the differences. So, based on the results, we recommended to manage two independent population in the text.

  1. The conservation/management suggestions are limited in this study. I recommend that the authors write the more specific conservation/management strategies based on your results.

Response: Thanks for your good suggestion. Based on our results, we think that Changes to the population structures within a stock assessment can have profound implications on the stock status and management actions. The impacts of uncertain population structures on the ability of management strategies to reach their stated goals can be examined by conducting management strategy evaluations [53]. In order to achieve sustainable fisheries usage, we recommend managing two independent blue shark populations in the Pacific Ocean (line 355-361). The recommendation we suggest is that managing as two independent population. As for the specific conservation/management strategies, it is hard to provide specific strategies. But we would like to performance MSE to predict the consequences when it manged as a signal population, two independent population and a meta-population in our future work.

Reviewer 4 Report

It is interesting to evaluate the manuscriptBlue shark (Prionace glauca) distribution in the Pacific Ocean: A look at continuity and size differences”. The work is interesting and broadens the knowledge about the Blue shark, and should intensive work need to get published for the conservation of Shark and other genetic resources of Oceans,  but the manuscript needs hard revision for publication. It has issues with the presentation of the work.

I have some comments here that the author must revise:

1.      Please see line no 4 …..why ‘and’ after all authors

2.      Abstract needs major revision focusing on the results and recommendation in brief.  

3.      Introduction section needs more review of the literature and hypothesis addressed in the manuscript. I suggest a more recent literature review, which needs to be incorporated to highlight areas for future study and call attention to research hypotheses. There are sentences “Northwest Atlantic Ocean [5], Pacific Ocean [6], and Mediterranean Sea 41 [7]” which need reframing.

4.      I also suggest the authors take reference to recently published works on Water and other Journals publishing aquatic resources.  Focused related to this study.

5.      Looking at the map, it is observed that the maximum population is restricted to some areas only, and therefore, the reason for maximum diversity should be mentioned in the article.

6.      Methodology section needs slight improvement. Methodology for Muscle tissue where collection techniques need to be highlighted

7.      There is some discussion in this section rather than a statement of the results of the analysis. The author must revise it carefully.

8.      The discussion section needs to go deeper and more systematically and add some references.

9.      Conclusion section is missing and therefore requests the authors to describe the specific conclusions obtained from this study. What are the advantages and disadvantages of this study, that should be focused on and should be freed from any references?

            I suggest major improvement of the manuscript before acceptance

Author Response

Comments and Suggestions for Authors

It is interesting to evaluate the manuscript “Blue shark (Prionace glauca) distribution in the Pacific Ocean: A look at continuity and size differences”. The work is interesting and broadens the knowledge about the Blue shark, and should intensive work need to get published for the conservation of Shark and other genetic resources of Oceans,  but the manuscript needs hard revision for publication. It has issues with the presentation of the work.

I have some comments here that the author must revise:

  1. Please see line no 4 …..why ‘and’ after all authors

Response: Thanks for your good suggestion. We have deleted and in the line 4 in the text.

  1. Abstract needs major revision focusing on the results and recommendation in brief.  

Response: Thanks for your good suggestion. We have rewritten the abstract as “ Blue shark (Prionace glauca) is a major bycatch species in the long-line and gill-net Pacific Ocean tuna fisheries, and the population structure is critical for fishery management. We employed generalized additive models to analyze the fork lengths of blue sharks and biological data (i.e., feeding level, sex, and genetic data), as well as environmental and spatial variables (i.e., sea surface temperature, month, longitude, and latitude) collected from 2011 to 2014 by the Chinese Thunnus alalunga long-line tuna fishery observer program. Fork length was significantly affected (P < 0.05) with location (latitude and longitude) and sex, and positively effected with sea surface temperature. No relationships were found between fork length and feeding level, month, and genetic data. We detected fork length differences among blue sharks over the range of the observed data, but the genetic data implied a panmictic population. Thus, we hypothesize that the genetic similarity was so close that it could not be well separated. Based on the precautionary principle, we recommend that the blue shark in the Pacific Ocean should be managed as two independent populations to ensure its sustainable use.”

  1. Introduction section needs more review of the literature and hypothesis addressed in the manuscript. I suggest a more recent literature review, which needs to be incorporated to highlight areas for future study and call attention to research hypotheses. There are sentences “Northwest Atlantic Ocean [5], Pacific Ocean [6], and Mediterranean Sea 41 [7]” which need reframing.

Response:Thanks for you good suggestion. We have revised the instruction as your suggestion.

  1. I also suggest the authors take reference to recently published works on Water and other Journals publishing aquatic resources.  Focused related to this study.

Response:Thanks for you good suggestion. We have revised the text as your suggestion.

  1. Looking at the map, it is observed that the maximum population is restricted to some areas only, and therefore, the reason for maximum diversity should be mentioned in the article.

Response:Thanks for you good suggestion. We have revised the text as your suggestion.

  1. Methodology section needs slight improvement. Methodology for Muscle tissue where collection techniques need to be highlighted

Response:Thanks for you good suggestion. We have revised as “Muscle tissue were collected under the back fin on the board from 98 individual blue sharks in three locations (denoted as Central Pacific Ocean Part A (CPA), Central Pacific Ocean Part B (CPB), and Central Pacific Ocean Part C (CPC)) of Pacific Ocean during 2011 to 2014. Among these individuals, 31, 34, and 33 were collected from the areas comprising 0–15N, 165–180W (CPA), 0–10S, 155–165W (CPB), and 0–10S, 145–155W (CPC), respectively (Figure 2).” in the text.

  1. There is some discussion in this section rather than a statement of the results of the analysis. The author must revise it carefully.

Response:Thanks for you good suggestion. We have revised the discussion as your suggestion.

  1. The discussion section needs to go deeper and more systematically and add some references.

Response:Thanks for you good suggestion. We have revised the discussion as your suggestion, as well as from other reviewers.

  1. Conclusion section is missing and therefore requests the authors to describe the specific conclusions obtained from this study. What are the advantages and disadvantages of this study, that should be focused on and should be freed from any references?

Response:Thanks for you good suggestion. We have added an conclusion “In this study, we used GAM to analyze the fork lengths of blue sharks and feeding level, sex, and genetic data, sea surface temperature, month, longitude, and latitude. The results indicated that fork length was significantly affected (P < 0.05) with location (latitude and longitude) and sex, and positively effected with sea surface temperature. But the relationship between fork length and feeding level, month, and genetic data are poor. And 98 individuals from 3 sampling sites were used to detected the genetic population in the studying areas, the results implied a panmictic population. Based on the analysis by synthesis, we hypothesize that the genetic similarity was so close that it could not be well separated. Based on the precautionary principle, we recommend that the blue shark in the Pacific Ocean should be managed as two independent populations to ensure its sustainable use.” in the text.

Round 2

Reviewer 1 Report

Manuscript is improved, but there are still several items that need clearing up. Here is an annotated list.

Line 40 - reword - 'calling for scientists to pay more attention to population...."

LIne 53 - 'concentrated' is awkward wording, maybe use 'found'

Line 64 - change 'stages' to 'Life stages...'

Line 71 - change 'know about the' to 'understand'

Line 75 - 'Our results would like to study...' is very awkward, not sure what authors mean here, needs rewriting

Line 92 - the 'And the shark.....' sentence belongs further down in methods, in the Biological data collected section.

Line 152. change to 'Variables included in....and their performance results'

Line 182 - unclear why you are referring to 'building the histogram' in this figure. the only histogram is the next figure. Does this info in this figure go into the next? can you do without this sentence? if you leave it, it needs to be clarified as to what exactly you mean.

Line 198 - in both x-axis label and fig. 6 title you have misspelled 'expected'

Line 215 - change to 'Effects of estimated.....(left), and the effect of sea surface....'

Line 356 - 'significantly correlated with....' would be better phrasing, i think

Line 357 - 'positively effected...' is wrong! do you mean positively correlated with or positively related to? give a p-value as you did above.....

Author Response

Line 40 - reword - 'calling for scientists to pay more attention to population...."

Response: Thanks for your suggestion. We have revised as your suggestion in the text.

LIne 53 - 'concentrated' is awkward wording, maybe use 'found'

Response: Thanks for your suggestion. We have revised as your suggestion in the text.

Line 64 - change 'stages' to 'Life stages...'

Response: Thanks for your suggestion. We have revised as your suggestion in the text.

Line 71 - change 'know about the' to 'understand'

Response: Thanks for your suggestion. We have revised as your suggestion in the text.

Line 75 - 'Our results would like to study...' is very awkward, not sure what authors mean here, needs rewriting

 Response: Thanks for your suggestion.  We have rewritten as “Our results would like to provide scientific recommendations for sustainable fisheries and appropriate management of the species. “ in the text.

Line 92 - the 'And the shark.....' sentence belongs further down in methods, in the Biological data collected section.

Response: Thanks for your suggestion. We have revised as your suggestion in the text.

Line 152. change to 'Variables included in....and their performance results'

Response: Thanks for your suggestion. We have revised as your suggestion in the text.

Line 182 - unclear why you are referring to 'building the histogram' in this figure. the only histogram is the next figure. Does this info in this figure go into the next? can you do without this sentence? if you leave it, it needs to be clarified as to what exactly you mean.

Response: Thanks for your suggestion.We deleted “We count the number of site differences between each pair of sequences in a sample, and use the resulting counts to build the histogram.” in the text.

Line 198 - in both x-axis label and fig. 6 title you have misspelled 'expected'

Response: Thanks for your suggestion.Figure 6. Test results for log(FL). The fit observed cummulative probability to excepted cummulative probability was used to test the distribution of log(FL) was approximately normal. ‘excepted’ were included in the text.

Line 215 - change to 'Effects of estimated.....(left), and the effect of sea surface....'

Response: Thanks for your suggestion. We have revised as your suggestion in the text.

Line 356 - 'significantly correlated with....' would be better phrasing, i think

Response: Thanks for your suggestion. We have revised as your suggestion in the text.

Line 357 - 'positively effected...' is wrong! do you mean positively correlated with or positively related to? give a p-value as you did above.....

Response: Thanks for your suggestion. And we have rewritten as “and the effected with sea surface temperature were analyzed.” in the text.

Reviewer 3 Report

I think that the authors appropriately respond to my comments. I recommend that the authors check the manuscript again to make sure if there are minor mistakes in the manuscript. Thank you for giving me the opportunity for reviewing this manuscript.

Author Response

I think that the authors appropriately respond to my comments. I recommend that the authors check the manuscript again to make sure if there are minor mistakes in the manuscript. Thank you for giving me the opportunity for reviewing this manuscript.

Response: Thanks for your suggestions and we improved the MS. In addition, we have revised as your suggestion in the text.  

Reviewer 4 Report

Can be accepted in current form

Author Response

Can be accepted in current form.

Response: Thanks for your suggestions and we improved the MS.
